# Relationship between diabetes self-care practices and control of periodontal disease among type2 diabetes patients in Bangladesh

**S. M. Mahmudul Hasan**[1], **Mosiur Rahman**[1,2], **Keiko Nakamura**[1]\*, **Yuri Tashiro**[1], **Ayano Miyashita**[1], **Kaoruko Seino**[1]

**1** Division of Public Health, Department of Global Health Entrepreneurship, Graduate School of Medical and Dental Sciences, Tokyo Medical and Dental University, Tokyo, Japan, **2** Department of Population Science and Human Resource Development, University of Rajshahi, Rajshahi, Bangladesh

\* nakamura.ith@tmd.ac.jp

## Abstract

### Introduction

The prevalence of periodontal disease is high in diabetes patients worldwide, including Bangladesh. Although associations of periodontal disease outcomes and clinical determinants of diabetes have been investigated, few studies have reported on the relationship between periodontal diseases outcomes with modifiable factors, such as self-care and oral hygiene practices, in patients with diabetes. Moreover, in order to develop targeted strategies, it is also important to estimate their aggregated contribution separately from that of the established sociodemographic and diabetics related clinical determinates. Therefore, this study was performed to elucidate 1) the relationship of diabetes patients' self-care and oral hygiene practices to periodontal disease and 2) the relative contributions of selected factors to periodontal disease outcome in type 2 diabetes patients.

### Methods

The data were obtained from the baseline survey of a multicentre, prospective cohort study. A total of 379 adult patients with type 2 diabetes from three diabetic centres in Dhaka, Rajshahi and Barishal, received periodontal examinations using the community periodontal index (CPI) probe, glycated haemoglobin examination, other clinical examinations, and structured questionnaires. Multiple logistics regression analyses were performed to assess the associations between selected factors and prevalence of any periodontal disease and its severity.

### Results

More than half of the participants were female (53.8%) and 66.8% of the total participants was 21–50 years old. The prevalence of any (CPI code 2+3+4; 75.7%) and severe form (CPI code 4; 35.1%) of periodontal disease were high in type 2 diabetes patients. In multivariate analysis, the odds of periodontal disease increased with unfavourable glycaemic control indicated by HbA1c $\geq$ 7%, and decreased by 64%, 85% and 92% with adherence to

**Data Availability Statement:** All relevant data are within the paper and its Supporting Information file.

**Funding:** KN received funds from by Grant-in-Aid for Scientific Research (17F17110 and 17H02164) of the Japan Society of the Promotion of Science (JSPS) [https://www.jsps.go.jp/english/index.html] and GACD Collaborative Call (SU12) of Japan Agency for Medical Research and Development (AMED) [https://www.amed.go.jp/en/index.html]. The funders had no role in study design, data collection and analysis, decision to publish, or preparation of the manuscript.

**Competing interests:** The authors have declared that no competing interests exist.

recommended diet, physical activity, and oral hygiene practices, respectively. Diabetes self-care practice explained the highest proportion of the variance (13.9%) followed by oral hygiene practices (10.9%) by modelling any periodontal disease versus no disease. Variables of diabetes conditions and oral hygiene practices explained 10.9% and 7.3% of the variance by modelling severe (CPI code 4) or moderate (CPI code 3) forms of periodontal disease versus mild form of periodontal disease. Findings also conferred that while poor diabetes control had an individually adverse association with any form of periodontal diseases and its severity, the risk of diseases was moderated by oral hygiene practices.

## Conclusions

This study suggested that, in addition to diabetes-related clinical determinants, self-care practices, and oral hygiene practices must be taken into consideration for prevention and control of periodontal disease in patients with diabetes.

## Introduction

Diabetes is a chronic metabolic disorder that is approaching epidemic proportions globally and Bangladesh shows the highest prevalence of diabetes worldwide (10.8%) [1]. There is evidence that, in addition to macrovascular and microvascular complications, patients with diabetes have high rates of several oral health conditions, such as dental caries, dry mouth, cheilitis, and tooth loss [2, 3]. Periodontal disease is the most prevalent of these oral health conditions and has been labelled "the sixth complication of diabetes" [4]. Several studies have shown both increased prevalence and severity of periodontal disease in diabetic compared with non-diabetic subjects [5–8]. In Bangladesh, a recent single-centre hospital-based study concluded that there was a higher prevalence of the periodontal disease among patients with diabetes (55.8%) [9].

Epidemiological studies of periodontal disease indicated that several factors are associated with periodontal disease, i.e., advanced age, male sex, obesity, inadequate oral hygiene practices, smoking, and low socioeconomic status [10–14]. These reports were based on different population subgroups, such as school [10] and university students [11], low-risk pregnant women [12], adolescents [13], and elderly [14] general populations. However, there have been very few studies focused on periodontal disease in diabetes patients [5–8], a group who at increased risk of periodontal disease.

Studies focusing on periodontal disease among diabetes patients have mostly been conducted in developed countries [7, 8]. The relationship between periodontal health and diabetes-related clinical determinants, such as HbA1c [5] or duration of diabetes [5] have been analysed. However, there have been few studies of the associations of periodontal health with modifiable factors, such as self-care practices, healthy diet, and physical activity, as well as oral hygiene-related knowledge, attitudes, and behaviours. Lack of self-care practices such as inadequate diet, and low physical activity may exacerbate the management of the disease among patients with diabetes, which can ultimately contribute to poor control of diabetes [15] and increase the risk of periodontal disease by incorporating inflammation, immune functioning, neutrophil activity, and cytokine biology [16].

While there is evidence of an understanding of various factors that affect the outcomes of periodontal disease, there is little understanding of their relative contribution to explaining variance in periodontal disease. The results of relative contribution of individual factors could

help policy makers making decisions for developing targeted strategies such as prioritizing interventions for better management of periodontal disease among diabetes patients. If, for example, self-care practices are found to explain the variance in periodontal disease outcome more than sociodemographic and other factors, then decision-makers may realize the advantages of adopting policies that enable more patients to adhere to prescribed diet, physical activity and others, hence reducing prevalence of periodontal disease.

Furthermore, factors related to periodontal disease among patients with diabetes may differ according to the developmental context, suggesting a need for research to discuss contextually appropriate strategies to mitigate periodontal disease burden by increased morbidity of diabetes in Bangladesh.

Therefore, this study was performed to examine periodontal disease in adult type 2 diabetes patients in Bangladesh. The objectives of this study were to elucidate 1) the relationship between periodontal disease and the following factors: [a] diabetes patients' self-care practices and [b] oral hygiene-related knowledge, attitudes and behaviours and 2) the relative contributions of selected factors to periodontal disease outcomes among type 2 diabetes patients.

## Methods

### Participants and recruitment

Data were obtained from the baseline survey of the multicentre, prospective cohort "Barriers to Glycemic Control and Periodontal Disease Study". The survey was conducted between August and September 2018 at outpatient departments of three Diabetic Associations of Bangladesh (DAB)-affiliated hospitals specialising in diabetes patients, located in Dhaka, Rajshahi, and Barishal. The inclusion criteria were type 2 diabetes patients aged 21–59 years, permanent residents in the study locations, diagnosed for diabetes at least 1 year previously, and with all index teeth necessary to conduct a valid periodontal examination. The exclusion criteria included previously received insulin prescriptions that may indicate more advanced stage of the disease, diagnosis with gestational diabetes, and prior inclusion in any pharmacologic clinical trials. Finally, 379 patients were selected from these three centres in equal proportions, i.e., 124 from Dhaka, 127 from Rajshahi, and 128 from Barisal.

### Sample size calculation

Assuming a rate of poor glycaemic control among type 2 diabetics with no severe periodontal disease of 84.7% [17], to achieve a power of 80%, a level of significance of 5%, and to detect a prevalence ratio of poor glycaemic control of 2.90 [17], for patients with no severe periodontal disease compared to patients with severe periodontal disease, the target sample size was 345. The sample size was further increased to 379 to allow for a 10% dropout rate.

### Data collection and study variables

The patients underwent periodontal disease examination by dentists, blood tests, blood pressure and anthropometric measurements as well as structured questionnaire interviews regarding sociodemographic characteristics, diabetes history, self-care practices, oral hygiene-related knowledge, attitudes, and behaviours.

### Periodontal disease examination

Periodontal disease condition was measured by Community Periodontal Index (CPI). Three trained dentists (one dentist per centre) conducted full-mouth examinations. All examinations were conducted in the diabetes centres. Examinations were performed with the patient seated

in an upright position in a dental chair under an overhead light. The all safety and biohazard measures were adopted by appropriately dressed examiners. Dental evaluations were performed in all sextants using a mouth mirror and Community Periodontal Index probe (CPI probe). The CPI probe was designed to assess periodontal status and proposed by the World Health Organisation (WHO) for use in recording CPI and is used worldwide for periodontal screening [18, 19]. The reliability of the CPI index in epidemiological assessment of periodontal diseases was described in detail elsewhere [20]. The examination took an average of 5–6 minutes. In each sextant, the index teeth were probed in six sites (mesio-buccal, mid-buccal, disto-buccal, disto-lingual, mid-lingual, and mesio-lingual). The index teeth (11, 16, 17, 26, 27, 31, 36, 37, 46, and 47) were examined. The scores adopted for CPI evaluation for individual index teeth were as follows: 0 = healthy periodontium; 1 = presence of bleeding on probing; 2 = presence of calculus; 3 = 4–5 mm pocket; 4 = 6 mm pocket or over. For each participant, the highest score on CPI reading among the six sextants was recorded as the final CPI score. Periodontal disease was defined as follows: CPI = 0, no disease; CPI = 1 or 2, mild; CPI = 3, moderate; CPI = 4, severe. We also created a binary variable to determine whether a patient was suffering from any form of periodontal disease (CPI $\geq$ 1).

## Haemoglobin A1c (HbA1c)

Venous blood was collected from patients between 08:00 and 11:00 for measurement of HbA1c, as an indicator of glycaemic control. After plasma separation, HbA1c was measured by high-performance liquid chromatography (D-10 (USA) HPLC Analyzer; Bio-Rad, Hercules, CA, USA). Glycaemic status was categorised as good (HbA1c $<$ 7%) and poor (HbA1c $\geq$ 7%) in accordance with the guidelines of the American Diabetes Association [21].

## Blood pressure and anthropometric measurements

Patients' blood pressure was measured using a blood pressure monitor (OMRON HAM-8731; Omron Healthcare, Kyoto, Japan). The means of three different systolic and diastolic blood pressure (BP) readings measured at 5-minute intervals were used. Hypertension was classified as systolic BP $\geq$ 140 mmHg or diastolic BP $\geq$ 90 mmHg or use of antihypertensive medication [22]. Body mass index (BMI) was calculated as weight (kg) divided by the square of height (m$^2$). Patients with BMI $\geq$ 25 were classified as overweight/obese [23].

## Questionnaire

The survey questionnaire was developed based on the WHO Oral Health Assessment for adults [24], combined with questions retrieved from a Demographic and Health Survey conducted in Bangladesh [25]. Additional questions regarding knowledge, attitudes, and behaviours toward oral health were developed through a review of the related literature [26] by the project staff and tested extensively in the field. The questionnaires were drafted in English, translated into Bangla, the national language of Bangladesh, the translations were reviewed by experts, and corrections were made according to a field pre-test.

**Sociodemographic characteristics.** Sociodemographic characteristics, including age, sex, education level, family structure type, marital status, area of residence, centre location, and socioeconomic status (SES) were considered for sociodemographic characteristics, were recorded. Variables were defined as follows: age, young adult (21–44 years), middle-aged adult (45–50 years) and older adult (51–59 years); sex, male vs. female; education level, no education (0 years), primary (1–5 years), secondary (6–9 years) or higher secondary or above (10 years or more); family structure type, nuclear vs. joint; currently married, yes vs. no; area of residence, rural vs. urban; centre location, Dhaka, Rajshahi or Barishal; and SES, low, medium or high.

SES was determined based on 26 selected household assets by principal component analysis and households were then divided into terciles according to the weighted wealth scores with each tercile designated a rank from 1 (poor) to 3 (rich).

**Diabetes history.** The duration of attending clinics for diabetes was classified as $< 5$ years or $\geq 5$ years.

**Self-care practices.** Dietary adherence questionnaires were adapted from the Summary of Diabetes Self-care Activities measure [27] and updated in compliance with the nutritional recommendations for adult Bangladeshi diabetics based on the recommended dietary regimens over the last 7 days [28]. A patient was considered as an adherent to the dietary recommendations if the calculated total score was $\geq 56$ (maximum score = 70). Participants' leisure-time physical activity was measured through the translated Bangla version [29] of the International Physical Activity Questionnaire (IPAQ). Patients were considered adherent to recommended physical activity if [moderate physical activity + vigorous physical activity $\times 2$] $\geq 150$ minutes in 7 days [30]. Participants were classified as tobacco users if they currently smoked any tobacco products, such as cigarettes, cigars, pipes or currently used any smokeless tobacco products, such as zarda, sadapata, gul, and snuff.

**Oral-hygiene practices.** Questions related to patients' knowledge related periodontal disease (6 questions), attitudes towards periodontal disease (6 questions), and oral hygiene behaviours (9 questions) were used. Scores for individual categories were developed, then terciles were used to grade patients' knowledge, attitude, and behavior as indicating poor, fair, or good.

Included questions were as follows:

[Knowledge] (1) periodontal disease can destroy teeth; (2) periodontal disease can affect overall health; (3) regular tooth brushing can recede onset of periodontal disease; (4) consumption of carbonated beverages can increase the risk of periodontal disease; (5) consumption of fruits and vegetables can prevent periodontal disease, and; (6) people with diabetes are more likely to have periodontal disease. Each correct response was awarded 1 point, while incorrect or "don't know" responses received no points, leading to a total possible score of 6 points.

[Attitudes] (1) it is important to take care of one's teeth; (2) it is important to visit a dentist regularly for periodontal disease; (3) regular and correct tooth brushing is necessary to prevent gum disease; (4) avoiding smoking is necessary to protect teeth from periodontal disease; (5) oral health examination is recommended for diabetic patients; and (6) eating a healthy diet is necessary to prevent periodontal disease. A score of 1 point was awarded for a "yes" response to each of statements 1–6 giving a total possible score of 6 points.

[Behaviours] (1) frequency of cleaning teeth per day; (2) brushing time, in minutes; (3) types of cleaning aids used; (4) materials used to clean teeth; (5) frequency of changing toothbrush; (6) types of toothpaste used; (7) mouth rinsing after eating; (8) frequency of tongue cleaning after meals or when brushing; and (9) frequency of eating candy/chocolate/sweets per day. A score of 1 was given for good oral health behaviours and a score of 0 was given for poor behaviours for a maximum score of 9 points.

Regarding internal consistency of the questionnaire on oral hygiene practices, Cronbach's α was 0.79. 0.78 and 0.81 for the knowledge, attitude, and behaviour instruments, respectively. Concerning reproducibility, the two sets of answers from the patients in the test-retest group were examined using the intraclass correlation coefficient. A coefficient $\geq 0.70$ was taken to indicate satisfactory test-retest reliability.

## Analyses

Prevalence of periodontal disease according to its severity in type 2 diabetes patients were calculated according to subjects' sociodemographic characteristics, diabetes-related clinical

determinants (duration of diabetes, HbA1c, hypertension, and overweight/obesity), self-care practices, and oral hygiene practices. Multivariable relationships between the measured variables and any periodontal disease categories versus no disease were explored by binary adjusted logistic regression analysis. Adjusted ordinal logistic analysis was conducted with mild disease as the reference level to explore the relationship between the selected risk factors for the severity of periodontal disease outcome, namely mild, moderate, and severe form of disease in type 2 diabetes patients. Brant's test [31] confirmed that the proportional odds assumption in our ordinal regression model was not violated. Independent variables were included simultaneously in the multiple regression models. The variance inflation factor was computed after each multivariable model to examine the presence of multicollinearity.

To measure the relative contributions of sociodemographic characteristics, diabetes-related clinical determinants, self-care practices, and oral hygiene practices to the prevalence of periodontal disease and its severity, we built hierarchical regression models and examined the coefficient of determination for logistic regression models (pseudo $R^2$) adjusted for the number of variables in the model. The initial model included the sociodemographic variables. Variables of diabetes-related clinical determinants, self-care practices, or oral hygiene practices were added to the model, individually or collectively. The increment in McFadden's pseudo $R^2$ (logistic models) was tested using Wald's $\chi^2$ test. In addition, a stratified analysis according to oral hygiene behaviours was performed to see if diabetes patients with poor or fair oral hygiene behaviours have a specific risk associated with any or severity type of periodontal disease, in terms of poor control of diabetes.

Statistical analyses were performed using Stata version 14 (StataCorp, College Station, TX). In all analyses, $P < 0.05$ (two-tailed) was taken to indicate statistical significance.

## Ethical aspects

The objectives and importance of the research were explained to all participants before recruitment into the study. Participation was voluntary and written informed consent was obtained from all participants. The study was approved by the Research Review Committee of Tokyo Medical and Dental University, Japan was granted an ethical clearance waiver from the research ethics boards of the Institute of Biological Sciences, University of Rajshahi, Bangladesh.

## Results

The sociodemographic, diabetes-related clinical determinants, self-care practices, and oral hygiene practice-related characteristics of the study sample are presented in Table 1. Among a total of 379 type 2 diabetes patients, 66.8% were aged 21–50 years old, 12.9% had no education, 82.3% lived in nuclear families, and 61.5% lived in rural areas. With respect to their marital status, 91.6% of the patients were currently married. Of the total sample population, 34.8% of the patients were classified as wealthy, 31.4% belonged to the middle wealth category, and 33.8% were defined as poor.

Table 1 also shows the prevalence of periodontal disease and its severity according to the sociodemographic characteristics, diabetes-related clinical determinants, self-care practices, and oral hygiene practice variables among diabetic patients. The rate of any periodontal disease in the 379 type 2 diabetes patients enrolled in this study was 75.7%; mild periodontal disease, 15.5%; moderate periodontal disease, 25.1%; and severe periodontal disease, 35.1%. Any and severe periodontal disease were significantly more prevalent among participants with a low level of education, low SES, and living in a nuclear family; those with diabetes duration $\geq$ 5years; those with HbA1c $\geq$ 7%; overweight or obese patients; non-adherence to the

**Table 1. Prevalence of any periodontal disease and its severity in type 2 diabetic patients according to sociodemographic characteristics, diabetes-related clinical determinants, self-care practices, and oral hygiene practices.**

| Characteristics | n (%) | Any periodontal disease[1] (n = 379) | | Periodontal disease severity level[2] (n = 287) | | | |
|---|---|---|---|---|---|---|---|
| | | % (95% CI) | P-value[3] | Mild (n = 59) % (95% CI) | Moderate (n = 95) % (95% CI) | Severe (n = 133) % (95% CI) | P-value[4] |
| **Sociodemographic characteristics** | | | | | | | |
| Age, years | | | | | | | |
| 21–44 | 119 (31.4) | 67.2 (58.2–75.1) | 0.003 | 18.7 (11.6–28.9) | 28.8 (19.8–39.7) | 52.5 (41.5–63.2) | 0.240 |
| 45–50 | 134 (35.4) | 73.9 (65.7–80.7) | | 20.2 (13.4–29.3) | 42.4 (33.0–52.4) | 37.4 (28.3–47.4) | |
| 51–59 | 126 (33.2) | 85.7 (78.4–90.8) | | 22.2 (15.3–31.1) | 27.8 (20.1–37.0) | 50.0 (40.6–59.4) | |
| Sex | | | | | | | |
| Female | 204 (53.8) | 68.1 (61.3–74.2) | <0.001 | 27.3 (20.5–35.4) | 27.3 (20.5–35.4) | 45.4 (37.2–53.7) | 0.172 |
| Male | 175 (46.2) | 84.6 (78.4–89.2) | | 14.2 (9.4–20.8) | 38.5 (31.0–46.6) | 47.3 (39.3–55.4) | |
| Education | | | | | | | |
| No education | 49 (12.9) | 91.8 (80.0–96.9) | 0.018 | 11.1 (4.6–24.3) | 28.9 (17.4–43.9) | 60.0 (45.0–73.3) | <0.001 |
| Primary | 129 (34.0) | 77.5 (69.5–83.9) | | 13.0 (7.6–21.2) | 25.0 (17.4–34.5) | 62.0 (52.0–71.0) | |
| Secondary | 99 (26.1) | 71.7 (62.0–80.0) | | 32.4 (22.5–44.2) | 35.2 (24.9–47.0) | 32.4 (22.5–44.2) | |
| Higher secondary and above | 102 (27.0) | 69.6 (60.0–77.8) | | 25.4 (16.5–36.8) | 45.1 (33.8–56.8) | 29.5 (20.1–41.3) | |
| Family structure type[5] | | | | | | | |
| Nuclear | 312 (82.3) | 77.9 (72.9–82.2) | 0.034 | 19.3 (14.6–24.8) | 31.3 (25.7–37.4) | 49.4 (43.1–55.7) | 0.022 |
| Joint | 67 (17.7) | 65.7 (53.4–76.1) | | 27.3 (16.0–42.4) | 43.2 (29.3–58.2) | 29.5 (17.8–44.7) | |
| Currently married | | | | | | | |
| No | 32 (8.4) | 87.5 (70.6–95.3) | 0.104 | 10.7 (3.4–29.0) | 17.9 (7.4–36.9) | 71.4 (51.9–85.3) | 0.009 |
| Yes | 347 (91.6) | 74.6 (70.0–79.0) | | 21.6 (17.0–27.1) | 34.7 (29.2–40.8) | 43.6 (37.7–49.8) | |
| Area of residence | | | | | | | |
| Rural | 146 (61.5) | 77.4 (70.0–83.5) | 0.548 | 21.2 (14.6–29.8) | 28.3 (20.7–37.4) | 50.4 (41.2–59.6) | 0.460 |
| Urban | 233 (38.5) | 74.7 (68.7–83.5) | | 20.1 (14.8–26.8) | 36.2 (29.4–43.6) | 43.7 (36.4–51.2) | |
| Centre | | | | | | | |
| Rajshahi | 127 (33.5) | 85.0 (77.7–90.3) | 0.001 | 19.4 (13.0–28.1) | 36.1 (27.5–44.5) | 44.4 (35.3–54.0) | 0.901 |
| Barishal | 128 (33.8) | 77.3 (69.2–83.8) | | 23.2 (15.9–32.6) | 27.3 (19.5–36.9) | 49.5 (39.7–59.3) | |
| Dhaka | 124 (33.7) | 64.5 (55.7–72.5) | | 18.8 (11.6–28.9) | 36.3 (26.4–47.4) | 45.0 (34.3–56.1) | |
| SES | | | | | | | |
| Low | 128 (33.8) | 85.2 (77.8–90.4) | <0.001 | 9.2 (5.0–16.3) | 21.1 (14.4–29.4) | 69.7 (60.4–77.7) | <0.001 |
| Medium | 119 (31.4) | 84.0 (76.3–89.6) | | 30.0 (21.8–39.8) | 34.0 (25.3–43.9) | 36.0 (27.1–45.9) | |
| High | 132 (34.8) | 59.1 (50.5–67.2) | | 24.4 (16.0–35.2) | 48.7 (37.8–59.8) | 26.7 (18.2–37.9) | |
| **Diabetes-related clinical determinants** | | | | | | | |

*(Continued)*

**Table 1.** (Continued)

| Characteristics | n (%) | Any periodontal disease[1] (n = 379) | | Periodontal disease severity level[2] (n = 287) | | | |
|---|---|---|---|---|---|---|---|
| | | % (95% CI) | P-value[3] | Mild (n = 59) % (95% CI) | Moderate (n = 95) % (95% CI) | Severe (n = 133) % (95% CI) | P-value[4] |
| Duration of diabetes | | | | | | | |
| <5 years | 197 (52.0) | 71.1 (64.3–77.0) | 0.028 | 25.7 (19.1–33.7) | 45.0 (36.9–53.4) | 29.3 (22.3–38.4) | <0.001 |
| ≥5 years | 182 (48.0) | 80.8 (74.5–85.9) | | 15.6 (10.6–22.5) | 21.8 (16.0–29.2) | 62.6 (54.4–70.1) | |
| HbA1c[6] | | | | | | | |
| <7% | 104 (27.4) | 54.8 (45.1–64.2) | <0.001 | 38.6 (26.8–51.9) | 50.9 (38.0–63.7) | 10.5 (4.8–21.7) | <0.001 |
| ≥7% | 275 (72.6) | 83.6 (78.8–87.6) | | 16.1 (11.9–21.5) | 28.7 (23.2–34.9) | 55.2 (48.7–61.6) | |
| Hypertension[7] | | | | | | | |
| No | 202 (53.3) | 75.2 (68.8–80.7) | 0.817 | 25.0 (18.7–32.5) | 37.5 (30.1–45.5) | 37.5 (30.1–45.5) | 0.001 |
| Yes | 177 (46.7) | 76.3 (69.4–82.0) | | 15.6 (10.3–22.7) | 28.1 (21.2–36.3) | 56.3 (47.8–64.5) | |
| Overweight/Obesity[8] | | | | | | | |
| No | 214 (56.5) | 68.2 (61.7–74.1) | <0.001 | 30.1 (23.2–38.2) | 41.1 (33.5–49.3) | 28.8 (22.0–36.7) | <0.001 |
| Yes | 165 (43.5) | 85.5 (79.2–90.1) | | 10.6 (6.5–17.0) | 24.8 (18.3–32.6) | 64.5 (56.2–72.0) | |
| **Self-care practices** | | | | | | | |
| Adherence to recommended diet | | | | | | | |
| No | 287 (75.7) | 84.0 (79.2–87.8) | <0.001 | 21.6 (16.8–27.3) | 27.8 (22.5–33.8) | 50.6 (44.2–56.9) | 0.038 |
| Yes | 92 (24.3) | 50.0 (39.8–60.2) | | 15.2 (7.3–28.9) | 60.9 (46.0–73.9) | 23.9 (13.6–38.5) | |
| Adherence to recommended physical activity | | | | | | | |
| No | 233 (61.5) | 86.3 (81.2–90.0) | <0.001 | 23.9 (18.4–30.4) | 24.8 (19.4–31.4) | 51.2 (44.3–58.1) | 0.306 |
| Yes | 146 (38.5) | 58.9 (50.7–66.6) | | 12.8 (7.2–21.7) | 52.3 (41.7–62.7) | 34.9 (25.5–45.6) | |
| Use of tobacco products[9] | | | | | | | |
| Yes | 47 (12.4) | 87.2 (74.1–94.2) | 0.049 | 17.1 (8.3–32.0) | 29.3 (17.4–45.1) | 53.7 (38.3–68.4) | 0.321 |
| No | 332 (87.6) | 74.1 (69.1–78.5) | | 21.1 (16.5–26.7) | 33.7 (28.1–39.9) | 45.1 (39.0–51.4) | |
| **Oral hygiene practices** | | | | | | | |
| Knowledge grading | | | | | | | |
| Poor | 140 (36.9) | 91.4 (85.4–95.1) | <0.001 | 11.7 (7.2–18.6) | 21.9 (15.5–29.9) | 66.4 (57.8–74.1) | <0.001 |
| Fair | 117 (30.9) | 77.8 (69.3–84.4) | | 24.2 (16.4–34.1) | 44.0 (34.0–54.3) | 31.9 (23.1–42.2) | |
| Good | 122 (32.2) | 55.7 (46.8–64.3) | | 32.4 (22.2–44.4) | 39.7 (28.7–51.8) | 27.9 (18.5–39.9) | |
| Attitude grading | | | | | | | |
| Poor | 127 (33.5) | 88.2 (81.3–92.8) | <0.001 | 8.9 (4.8–15.9) | 27.7 (20.1–36.8) | 63.4 (54.0–71.8) | <0.001 |
| Fair | 128 (33.8) | 79.7 (71.8–85.8) | | 23.5 (16.2–32.8) | 33.3 (24.8–43.1) | 43.1 (33.8–53.0) | |
| Good | 124 (32.7) | 58.9 (50.0–67.2) | | 32.2 (24.2–45.9) | 41.1 (30.3–52.8) | 24.7 (16.0–35.9) | |

*(Continued)*

**Table 1.** (Continued)

| Characteristics | n (%) | Any periodontal disease[1] (n = 379) | | Periodontal disease severity level[2] (n = 287) | | | |
|---|---|---|---|---|---|---|---|
| | | % (95% CI) | P-value[3] | Mild (n = 59) % (95% CI) | Moderate (n = 95) % (95% CI) | Severe (n = 133) % (95% CI) | P-value[4] |
| Behaviour grading | | | | | | | |
| Poor | 139 (36.7) | 92.8 (87.1–96.1) | <0.001 | 10.1 (5.9–16.6) | 23.3 (16.7–30.6) | 66.7 (58.0–74.3) | <0.001 |
| Fair | 105 (27.7) | 81.0 (72.3–87.4) | | 22.4 (14.7–32.5) | 40.0 (30.1–50.8) | 37.6 (27.9–48.4) | |
| Good | 135 (35.6) | 54.1 (45.6–62.3) | | 37.0 (26.6–46.7) | 42.5 (31.6–54.1) | 20.5 (12.7–31.5) | |
| **Percentage among all diabetes patients participated[10]** | | 75.7 (71.4–80.1) | | 15.5 (12.2–19.6) | 25.1 (20.9–29.7) | 35.1 (30.4–40.1) | |
| **Percentage among diabetes patients with any periodontal disease[11]** | | – | | 20.5 (16.2–25.6) | 33.1 (27.8–38.7) | 46.4 (40.6–52.2) | |

Notes: CI: confidence interval

[1]comparison was made for all diabetes patients with any periodontal disease categories versus. no disease

[2]comparison among mild, moderate, and severe category of patients diagnosed with periodontal disease

[3]$\chi^2$ test

[4]$\chi^2$ test for trend was performed for ordinal outcomes (mild, moderate, and severe periodontal disease categories) and ordinal independent variables and the Mann–Whitney U test was performed for ordinal outcomes (mild, moderate, and severe periodontal disease categories) and binary independent variables

[5]Nuclear family: a family group consisting only of parents and children; joint family: where more than one generation live together in a common house

[6]HbA1c < 7% was regarded a good control of diabetes

[7]Defined by systolic blood pressure (SBP) $\geq$ 140 mmHg or diastolic blood pressure (DBP) $\geq$ 90 mmHg

[8] Defined by BMI $\geq$ 25 kg/m$^2$

[9]Current smoker or user of smokeless tobacco products, such as tobacco leaf, zarda, gul, etc

[10]calculated for all diabetes patients with any type of periodontal disease versus no disease;

[11] calculated for mild, moderate, and severe categories of diagnosed periodontal disease patients except no category of disease

recommended diet for diabetes patients; and poor knowledge, attitude, and behaviours toward oral hygiene.

S1 Table shows the adjusted odds ratio (AOR) for associations between various measures of oral hygiene practices according to the prevalence of any periodontal disease among type 2 diabetic patients. Selected knowledge and attitudes variables independently significantly associated with lower odds of a diagnosis of any periodontal disease. Selected oral health-related behaviours, such as brushing time $\geq$ 2 minutes, use of fluoridated toothpaste, and mouth rinsing after eating, resulted in reductions of 0.44, 0.18 and 0.25 times in likelihood of any periodontal disease, respectively.

Table 2 shows the AOR of the likelihood of a diagnosis of any periodontal disease and its severity among diabetes patients according to sociodemographic characteristics, diabetes-related clinical determinants, self-care practices, and oral hygiene practice variables. Regarding diabetes-related clinical determinants, patients with HbA1c $\geq$ 7% suggesting poor control of diabetes were 2.73 times more likely to exhibit periodontal disease. Longer duration of diabetes $\geq$ 5 years, HbA1c $\geq$ 7%, hypertension, and high BMI indicating overweight or obesity were significantly associated with a diagnosis of severe periodontal disease. Since the proportional odds assumption was held, the interpretations of the results obtained by modeling moderate periodontal disease versus mild were the same. With regard to self-care practices, adherence to the recommended diet was associated with both reductions in odds of a diagnosis of periodontal disease and severe periodontal disease. Adherence to physical activity was associated with

**Table 2. Adjusted odds ratios (AOR) for associations between sociodemographic characteristics, diabetes-related clinical determinants, self-care practices, and oral hygiene practices according to periodontal disease and its severity among individuals with type 2 diabetes.**

| Characteristics | Any periodontal disease[1] (n = 379) | | Periodontal disease severity level (n = 287) | |
|---|---|---|---|---|
| | AOR[1] (95% CI) | *P*-value | AOR[2] (95% CI) | *P*-value |
| **Sociodemographic characteristics** | | | | |
| Age, years | | | | |
| 21–44 | 1.00 | - - | 1.00 | - - |
| 45–50 | 1.65 (0.73–3.73) | 0.228 | 1.91 (1.01–3.72) | 0.049 |
| 51–59 | 3.32 (1.34–8.19) | 0.009 | 1.41 (0.76–2.62) | 0.276 |
| Sex | | | | |
| Female | 1.00 | - - | 1.00 | - - |
| Male | 3.28 (1.46–7.37) | 0.004 | 1.75 (0.96–3.19) | 0.069 |
| Education | | | | |
| No education | 1.00 | - - | 1.00 | - - |
| Primary | 0.91 (0.21–4.03) | 0.902 | 1.16 (0.50–2.68) | 0.729 |
| Secondary | 1.56 (0.34–7.21) | 0.571 | 1.06 (0.42–2.65) | 0.898 |
| Higher secondary and above | 2.88 (0.54–15.22) | 0.214 | 1.32 (0.47–3.75) | 0.601 |
| Family structure type | | | | |
| Nuclear | 1.00 | - - | 1.00 | - - |
| Joint | 0.69 (0.29–1.63) | 0.392 | 0.82 (0.41–1.66) | 0.587 |
| Currently married | | | | |
| No | 1.00 | - - | 1.00 | - - |
| Yes | 0.49 (0.11–2.14) | 0.342 | 0.34 (0.11–1.05) | 0.060 |
| Area of residence | | | | |
| Rural | 1.00 | - - | 1.00 | - - |
| Urban | 0.78 (0.35–1.76) | 0.553 | 1.22 (0.66–2.24) | 0.533 |
| Centre | | | | |
| Rajshahi | 1.00 | - - | 1.00 | - - |
| Barishal | 0.60 (0.24–1.50) | 0.274 | 0.64 (0.33–1.24) | 0.189 |
| Dhaka | 0.31 (0.13–0.76) | 0.011 | 0.73 (0.36–1.45) | 0.364 |
| SES | | | | |
| Low | 1.00 | - - | 1.00 | - - |
| Medium | 1.11 (0.37–3.27) | 0.853 | 0.43 (0.20–0.90) | 0.025 |
| High | 0.29 (0.10–0.81) | 0.019 | 0.26 (0.11–0.58) | 0.001 |
| **Diabetes-related clinical determinants** | | | | |
| Duration of diabetes | | | | |
| <5 years | 1.00 | - - | 1.00 | - - |
| ≥5 years | 1.22 (0.58–2.54) | 0.603 | 2.46 (1.39–4.4.36) | 0.002 |
| HbA1c[5] | | | | |
| <7% | 1.00 | - - | 1.00 | - - |
| ≥7% | 2.73 (1.19–6.27) | 0.018 | 3.64 (1.82–7.26) | <0.001 |
| Hypertension | | | | |
| No | 1.00 | - - | 1.00 | - - |
| Yes | 1.62 (0.80–3.30) | 0.183 | 2.03 (1.19–3.46) | 0.009 |
| Overweight/Obesity | | | | |
| No | 1.00 | - - | 1.00 | - - |
| Yes | 1.89 (0.90–3.98) | 0.093 | 2.86 (1.63–5.00) | <0.001 |
| **Self-care practices** | | | | |

*(Continued)*

**Table 2.** (Continued)

| Characteristics | Any periodontal disease[1] (n = 379) | | Periodontal disease severity level (n = 287) | |
|---|---|---|---|---|
| | AOR[1] (95% CI) | P-value | AOR[2] (95% CI) | P-value |
| Adherence to recommended diet | | | | |
| No | 1.00 | -- | 1.00 | -- |
| Yes | 0.36 (0.16–0.81) | 0.014 | 0.47 (0.22–0.98) | 0.047 |
| Adherence to recommended physical activity | | | | |
| No | 1.00 | -- | 1.00 | -- |
| Yes | 0.15 (0.07–0.33) | <0.001 | 0.86 (0.48–1.53) | 0.601 |
| Use of tobacco products | | | | |
| Yes | 1.00 | -- | 1.00 | -- |
| No | 0.83 (0.22–3.05) | 0.775 | 0.76 (0.34–1.68) | 0.492 |
| **Oral hygiene practices** | | | | |
| Knowledge grading | | | | |
| Poor | 1.00 | -- | 1.00 | -- |
| Fair | 0.21 (0.06–0.78) | 0.020 | 0.38 (0.18–0.81) | 0.011 |
| Good | 0.50 (0.15–1.73) | 0.275 | 0.66 (0.26–1.64) | 0.372 |
| Attitude grading | | | | |
| Poor | 1.00 | -- | 1.00 | -- |
| Fair | 1.06 (0.34–3.25) | 0.825 | 1.05 (0.51–2.17) | 0.900 |
| Good | 1.15 (0.36–3.70) | 0.820 | 0.92 (0.37–2.29) | 0.853 |
| Behaviour grading | | | | |
| Poor | 1.00 | -- | 1.00 | -- |
| Fair | 0.88 (0.27–2.91) | 0.835 | 0.56 (0.26–1.21) | 0.137 |
| Good | 0.08 (0.02–0.29) | <0.001 | 0.19 (0.07–0.51) | 0.001 |

[1]Binary logistic regression analysiswas run for all diabetes patients with any periodontal disease categories versus no disease

[2]Ordinal logistic regression analysis was run for severity of periodontal disease-diagnosed patients, namely mild, moderate and severe patients with mild disease as the reference level

reductions in the odds of a diagnosis of periodontal disease. With regard to oral hygiene practices, fair knowledge about periodontal disease risk and good oral hygiene practices are associated with a reduction in the odds of a diagnosis of the disease and severe periodontal disease.

Table 3 shows estimates of the relative contributions of sociodemographic, diabetes-related clinical determinants, self-care practices, and oral hygiene practices to the prevalence of any periodontal disease and its severity. The initial models included only sociodemographic factors. The pseudo $R^2$ values for these variables were 0.187 for any form of periodontal disease and 0.096 for severe periodontal disease. In the models to add diabetes-related clinical determinants, self-care practices, and oral hygiene practices, independently, on sociodemographic factors, pseudo $R^2$ values for any form of periodontal disease increased by 0.069, 0.139, and 0.109, respectively. In the models to add diabetes-related clinical determinants, self-care practices, and oral hygiene practices, independently, on sociodemographic factors, pseudo $R^2$ values for the severe periodontal disease increased by 0.109, 0.004, and 0.073, respectively. The final model included all four groups of variables and the final pseudo $R^2$ values for this model were 0.444 for any form of periodontal disease and 0.262 for severe periodontal disease.

**Table 3. Hierarchical regression models of any periodontal disease and its severity among type 2 diabetics.**

| Model | Any periodontal disease[1] (n = 379) | | Periodontal disease severity level[2] (n = 287) | |
|---|---|---|---|---|
| | *pseudo-R²* | *P‡* | *pseudo-R²* | *P‡* |
| Sociodemographic characteristics | 0.187 | - - - | 0.096 | - - |
| Sociodemographic characteristics + diabetes-related clinical determinants | 0.256 | <0.001 | 0.205 | <0.001 |
| Sociodemographic characteristics + self-care practices | 0.326 | <0.001 | 0.100 | <0.001 |
| Sociodemographic characteristics + oral hygiene practices | 0.296 | <0.001 | 0.169 | <0.001 |
| Sociodemographic characteristics + diabetes-related clinical determinants + self-care practices + oral hygiene practices | 0.444 | <0.001 | 0.262 | <0.001 |

**Note:** Sociodemographic characteristics: age, sex, education, family structure, marital status, centre, area of residence and SES; diabetes-related clinical determinants: duration of diabetes, HbA1c, hypertension and overweight/obesity; self-care practices: adherence to recommended physical activity, adherence to recommended diet, use of tobacco products; oral hygiene practices: knowledge, attitude and practice grading related to oral hygiene behaviour.

‡ Statistical significance of the increase in pseudo-$R^2$ from the baseline model (sociodemographic characteristics alone).

[1] Binary logistic regression analysiswas run for all diabetes patients with any periodontal disease categories vs. no

[2] Ordinal logistic regression analysis was run for severity of periodontal disease-diagnosed patients, namely mild, moderate and severe patients with mild disease as the reference level

Table 4 shows the results of adjusted binary and ordinary logistic regression models for the association between any form of periodontal disease or its severity and control of diabetes interpreted by HbA1c measurements, after stratifying into three groups by oral hygiene behaviours. Patients with poor or fair oral hygiene practices with poor control of diabetes are associated with increased in the odds of a diagnosis of any and severe periodontal disease.

## Discussion

To our knowledge, this is the first detailed report regarding the prevalence of any form of periodontal disease and its severity and a wide range of sociodemographic, diabetes-self-care practice- and oral hygiene-related risk factors among patients with type 2 diabetes. This study yielded four main findings. First, type 2 diabetes patients in Bangladesh showed alarmingly high prevalence rates of both any (75.7%) and severe (35.1%) periodontal disease. Second, both

**Table 4. Association between periodontal disease and poor diabetes control stratified by oral hygiene behaviours.**

| Indicators and characteristics | Oral hygiene behaviours | | |
|---|---|---|---|
| | Poor | Fair | Good |
| Any periodontal disease among all diabetes patients[1] (n = 379) | n = 139 | n = 105 | n = 135 |
| HbA1c | | | |
| <7% | 1.00 | 1.00 | 1.00 |
| ≥7% | 28.95 | 44.62 | 1.95 |
| | (2.94–285.44)** | (5.19–383.3)** | (0.78–4.92) |
| Periodontal disease severity level among diabetes patients with any periodontal disease[2] (n = 287) | n = 129 | n = 85 | n = 73 |
| HbA1c | | | |
| <7% | 1.00 | 1.00 | 1.00 |
| ≥7% | 4.48 | 4.83 | 2.64 |
| | (1.32–15.20)* | (1.74–13.39)** | (0.80–8.76) |

[1] Binary logistic regression analysiswas run for any periodontal disease categories vs. no

[2] Ordinal logistic regression analysis was run for severity of periodontal disease-diagnosed patients, namely mild, moderate and severe patients with mild disease as the reference level. Here *, ** indicate $P<0/05$ and $P<0.01$

Models were adjusted by age, sex, education, family structure, marital status, centre, area of residence and SES

sociodemographic (e.g., higher age, low SES) and oral hygiene-related practice factors (e.g., fair knowledge about periodontal disease and adherence to good oral hygiene practices) play essential roles in the prevalence of any form of periodontal disease and its severity in diabetes patients as they do in the general population. Third, other than sociodemographic and oral hygiene-related practices, most of the variability in the prevalence of any form of periodontal disease and its severity were explained by diabetes self-care practices and diabetes-related clinical determinants, respectively. Therefore, in addition to sociodemographic and oral hygiene-related clinical determinants, diabetes-related clinical determinants and self-care practices should be taken into consideration for the prevention and control of periodontal disease in diabetes patients. Fourth, while poor diabetes control had an individually adverse association with any and severe form of periodontal diseases, the risk of diseases was moderated by oral hygiene practices.

The prevalence of any and severe form of periodontal disease in this study were higher than those reported in other single-centre hospital-based study in diabetes patients in Bangladesh using the Ramfjord periodontal index (75.7% vs. 55.8% and 35.1% vs. 30.9%, respectively) [9], and also than the values reported for the general Bangladeshi population in studies using the CPI (15% – 42.0% and 26.0%, respectively) [32, 33]. The heterogeneity of our findings from previous studies in Bangladesh can be accounted for by multiple elements, such as the selection of the general adult population other than diabetes patients or the variations in the available periodontal disease measures. In comparison with the general population, exceptionally high prevalence rates of any and severe periodontal disease in patients with type 2 diabetes were also reported in other countries using the same instrument (i.e., CPI) [6, 8].

As the high prevalence of diabetes has become an urgent public health concern in Bangladesh, this high prevalence of periodontal disease among diabetes may lead to poor quality of life in the absence of appropriate control and prevention strategies. Nearly 41% of cases of periodontal disease in this study were found to have mild or moderate disease, indicating a need for timely oral health care to avoid further progression. Although the Bangladeshi guidelines for type 2 diabetes discuss the control and prevention of five complications of diabetes [28], the results of this study suggested that more attention should be paid to periodontal disease among type 2 diabetes patients in Bangladesh.

Among sociodemographic characteristics, advancing age is a significant contributor to the extent of periodontal disease in the population. Significant differences in the extent and severity of periodontal disease were observed between young and old individuals in both the general population [34, 35] and patients with diabetes [6, 8]. This could be due to the general deterioration of immune function and tissue integrity with advancing age, resulting in increased susceptibility to periodontal disease. Higher prevalence rates of any and severe periodontal disease among men than women were reported in this study, as other studies reported [35, 36]. Groups with low SES showed a higher risk of any as well as a severe disease than groups with high SES, as other studies showed socioeconomic gradients were associated with the periodontal condition [35, 37].

The most important findings of this study were that other than sociodemographic risk factors most of the variability in the prevalence of any periodontal disease could be explained by diabetes self-care practices, which accounted for 13.9% of the observed variance. This study showed that self-care practices, such as adherence to recommended diet and physical activity, were associated with a lower likelihood of a diagnosis of periodontal disease. Good compliance with self-management of diabetes results in not only good control of HbA1c [38, 39] but also relate to better dental self-efficacy [38]. We performed additional analyses to confirm this hypothesis and the results suggested that patients with better adherence to the recommended diet and physical activity had higher rates of good oral hygiene practices as well as good glycaemic control.

Another important finding of this study was that, other than sociodemographic risk factors, most of the variability in the prevalence of severe periodontal disease could be explained by diabetes-related clinical determinants, which accounted for 10.9% of the observed variance. Consistent with the few studies reported to date in patients with type 2 diabetes [38, 39], factors related to diabetes condition, such as longer duration of diabetes ($\geq$ 5years), unfavourable glycaemic control (HbA1c $\geq$ 7%), hypertension, and overweight/obesity, were associated with higher likelihood of a diagnosis of severe periodontal disease.

This research indicates that both self-care practices and oral hygiene practices contribute substantially to the periodontal disease outcomes, in addition to the diabetes-related clinical determinants. For diabetes patients, the results provide implications for the prevention and control of the periodontal disease. The results also suggest to both medical and dental clinicians' as well as to policy makers', importance of collaborative approach encouraging diabetes patients to be compliant to both self-care and oral hygiene practices.

Our results also conferred that, although poor control of diabetes had an independently adverse association with any and severe form of periodontal disease, oral hygiene practices moderated the higher risk of the disease; patients who practiced poor or fair oral hygiene behaviours and poorly controlled diabetes were associated with more risk of developing severe form of the disease, than patients who practiced good oral hygiene behaviours and poorly controlled diabetes. When patients perform poor or fair oral hygiene practices, poor diabetes control adversely increases the risk of any and severe types of periodontal disease. The importance of this outcome must, therefore, be emphasized.

Previous studies have found that tobacco use affects even healthy individuals' periodontium [40], but for a smoker who has diabetes, the damage is multiplied [41]. In comparison, we found in the present study that the risk of any or severe periodontal disease was not correlated with tobacco use. The explanation for the insignificant correlation in our sample may be because more than half of the participants in our study were female and the prevalence of tobacco use among them is much lower compared to male consumption in Bangladesh. However, since the use of tobacco is a significant determinant of periodontal health, it is recommended that tobacco cessation and control of diabetes along with oral hygiene advice should be promoted as an integral component of periodontal therapy.

The results of this study showed that diabetes patients with fair knowledge regarding periodontal risk factors and good oral hygiene practices were less likely to suffer from any or severe periodontal disease. With regard to independent measures, consistent with studies in the general population [42, 43], our results indicated that level of knowledge of proper oral health (e.g., regular tooth brushing can reduce periodontal disease and consumption of carbonated beverages can increase the risk) or hygiene practices (e.g., brushing $\geq$ 2 minutes, rinsing one's mouth after eating and the use of fluoridated toothpaste) were associated with reduced prevalence of periodontal disease. Appropriate knowledge and good oral hygiene practices regarding periodontal health among diabetic patients are essential for better oral health outcomes. Interventions to diabetes patients should include encouragement to improve oral hygiene practices.

Interestingly, regardless of the negative effects of lack of appropriate knowledge and lack of good oral hygiene on periodontal health, we observed poor knowledge and hygiene practices in many of the measures of oral health (e.g., that regular tooth brushing can reduce periodontal disease or practices, such as correct brushing time or use of fluoridated toothpaste) among individuals with diabetes (S1 Table). This lack of oral health knowledge and hygiene practices may be attributable to a lack of institutionally based comprehensive oral and diabetes health care in Bangladesh.

Due to the lack of comprehensive care, diabetic patients may not obtain advice from dental practitioners regarding oral health care related to their diabetes. On the other hand, health care providers responsible for treating diabetes do not address the oral health care needs of their patients. A study reported that approximately 70% of diabetic patients had never received any advice from their dental practitioners about oral health care related to their diabetes [44]. Another study showed that the majority of health care providers who are responsible for treating diabetes did not discuss oral health care with their patients [45]. According to the need for co-management of oral and diabetes treatment with diabetes patients, a team effort involving the patient and both diabetes health care and dental practitioners is required.

This study had some limitations. First, there was a possibility of selection bias as the study was conducted only among participants visiting the outpatient departments of diabetes hospitals located in urban areas and therefore, they cannot be considered as a truly representative sample of the diabetic population in Bangladesh, thus limiting the generalisability of the results regarding the prevalence of periodontal disease. Further studies in patients in the community are therefore warranted. Second, data for several variables, such as adherence to the recommended diet, physical activity, and oral hygiene practices, were obtained by self-report and therefore some recall bias is concerned. However, self-reporting methods is understood to provide reasonably accurate estimates of adherence to diabetes self-management and oral hygiene practices [46].

Third, type 2 diabetes patients without using insulin were our study subjects. Insulin patients are normally on longer-term therapy, which is a risk factor for non-compliance with self-care behaviors and poor glycemic and periodontal disease control. Although the prevalence of insulin monotherapy among patients with type 2 diabetes in Bangladesh accounts for less than 7.3% to 14.0% [47, 48], additional research is needed to provide evidence-based guidelines for long-term diabetes patients in Bangladesh to keep control of the periodontal disease.

Finally, the cross-sectional nature of the study did not allow longitudinal assessment of the risk factors or causal relationship. Further longitudinal studies may overcome these limitations.

This study also had several strengths. First, this was a study of periodontal disease in systematically recruited diabetes patients based on examinations by trained dentists. Second, dental examinations were performed using the standard full-mouth examination. Third, the analysis was based on not only dental examinations but also the results of HbA1c, glycaemic control parameters obtained by blood examinations, physiological and anthropometric measurements. Fourth, the study involved a comprehensive analysis of dental and medical clinical parameters, behavioural and sociodemographic parameters. Finally, the random selection of multicentre data indicated the high degree of external validity of this study.

## Conclusions

The high prevalence of any and severe form of periodontal disease among patients with type 2 diabetes in Bangladesh is alarming. In addition to diabetes-related clinical determinants, self-care practices recommended for diabetes control and oral hygiene practices were related to the diagnosis of periodontal disease. Moreover, while poor diabetes control had an individually adverse association with any and severe form of periodontal diseases, the risk of diseases was moderated by oral hygiene practices. A comprehensive approach involving both diabetes and oral health practitioners is required for the prevention and control of periodontal disease in patients with diabetes.

## Supporting information

**S1 Table. Adjusted odds ratio (AOR) for associations between measures of oral hygiene practices according to prevalence of periodontal disease among type 2 diabetics individuals.**
(DOCX)

**S1 Data.**
(SAV)

## Acknowledgments

The authors gratefully acknowledge the fieldworkers performing the interviews and all the participating patients.

## Author Contributions

**Conceptualization:** S. M. Mahmudul Hasan, Mosiur Rahman, Keiko Nakamura.

**Data curation:** S. M. Mahmudul Hasan, Mosiur Rahman.

**Formal analysis:** S. M. Mahmudul Hasan, Mosiur Rahman, Keiko Nakamura, Yuri Tashiro.

**Funding acquisition:** Keiko Nakamura.

**Investigation:** S. M. Mahmudul Hasan, Mosiur Rahman, Keiko Nakamura.

**Methodology:** S. M. Mahmudul Hasan, Mosiur Rahman, Keiko Nakamura.

**Project administration:** S. M. Mahmudul Hasan, Mosiur Rahman, Keiko Nakamura, Ayano Miyashita, Kaoruko Seino.

**Resources:** Mosiur Rahman, Keiko Nakamura.

**Software:** Keiko Nakamura.

**Supervision:** Keiko Nakamura.

**Validation:** Keiko Nakamura, Yuri Tashiro, Kaoruko Seino.

**Writing – original draft:** S. M. Mahmudul Hasan, Mosiur Rahman, Keiko Nakamura.

**Writing – review & editing:** S. M. Mahmudul Hasan, Mosiur Rahman, Keiko Nakamura, Yuri Tashiro, Ayano Miyashita, Kaoruko Seino.

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
