## [Decision Letter · Decision Letter 0]

8 Jan 2021

PONE-D-20-36828

Relationship between diabetes self-care practices and control of periodontal disease among type 2 diabetes patients in Bangladesh

PLOS ONE

Dear Dr. Nakamura,

Thank you for submitting your manuscript to PLOS ONE. After careful consideration, we feel that it has merit but does not fully meet PLOS ONE’s publication criteria as it currently stands. Therefore, we invite you to submit a revised version of the manuscript that addresses the points raised during the review process.

We look forward to receiving your revised manuscript.

Kind regards,

Toshiyuki Ojima, M.D., Ph.D

Academic Editor

PLOS ONE

Journal Requirements:

2. Please include additional information regarding the survey or questionnaire used in the study and ensure that you have provided sufficient details that others could replicate the analyses.

For instance, if you developed a questionnaire as part of this study and it is not under a copyright more restrictive than CC-BY, please include a copy, in both the original language and English, as Supporting Information. Moreover, please include more details on how the questionnaire was pre-tested, and whether it was validated.

Reviewers' comments:

Reviewer's Responses to Questions

**Comments to the Author**

1. Is the manuscript technically sound, and do the data support the conclusions?

Reviewer #1: Yes

Reviewer #2: Yes

2. Has the statistical analysis been performed appropriately and rigorously? 

Reviewer #1: Yes

Reviewer #2: I Don't Know

3. Have the authors made all data underlying the findings in their manuscript fully available?

Reviewer #1: No

Reviewer #2: No

4. Is the manuscript presented in an intelligible fashion and written in standard English?

Reviewer #1: Yes

Reviewer #2: Yes

5. Review Comments to the Author

Reviewer #1: The paper was well written, reporting the prevalence and severity of periodontal disease, analyzing the relative contributions of a wide range of sociodemographic, diabetes self-care practice and oral hygienic factors among patients with type 2 diabetes in Bangladesh.

There are several minor problems need to be clarified.

Page 6, line 96. The exclusion criteria included previously receiving insulin prescriptions. Could you provide the percentage of diabetes patients who have received insulin prescriptions in Bangladesh? And, could you explain the reasons of such exclusion.

Table 1. What the difference between ”all” and “overall”? It’s confusing.

Table 1-3. The sample size for testing any form of periodontal disease was 379, but the sample size for testing the severity of periodontal disease was 287. Please explain the reasons in Methods, and make the columns of the tables more understandable.

For the general population of non-diabetic patients, poor oral hygiene habits are also an important risk factor for early occurrence of periodontal disease. In this study, oral health habits and glycemic control behaviors were put into a model at the same time in patients with diabetes mellitus. The results showed that both the glycemic control behaviors and the oral hygiene behaviors played an independent role in the development and severity of periodontal disease in diabetic patients. Could you further analyze the interaction between these two, and stratify by oral hygiene behavior to see if poor glycemic control is associated with much higher risks of having any forms or severity of periodontal disease among patients with poor oral hygiene?

Reviewer #2: This is a well written manuscript on an important subject area. However, the following suggestions has been made for improving the quality of the paper and better clarification.

Abstract: The second sentence of the Introduction need to be simplified, as it is difficult to understand in the current version. The justification of the second objective “) the relative contributions of selected factors to periodontal disease outcome in type 2 diabetes patients.” should be added.

Results: The Demographics of the participants need to be added (% of male, female, mean age (Standard deviation). The severity % is not clear in the sentence: “The prevalence (75.7%) and severity (35.1%) of periodontal disease were high in type 2 diabetes patients.” The last sentence of the Result “Diabetes conditions and oral hygiene practices explained 10.9% and 7.3% of the variance in severe periodontal disease.” The “severe periodontal disease” is not clear.

Introduction: The last sentence of Page 4 (line 71, 72) Association of periodontal health with self-care practice, healthy diet and physical activity has been shown as a gap in the literature. The potential undying mechanism of these associations need to be indicated to make the rational for investigating this associations. These factors seems to be directly related with the control of diabetes, which can have impact on periodontal health. (similar is the case for Discussion page 22, line 375 and 376. The other potential explanation should be added)

Another point, why it is important to estimate the relative contributions of individual factors to periodontal disease outcome need to be justified (clinical practice related importance etc.)

Results:

The demographic information of the study population need to be included in the first paragraph.

Discussion:

The clinical implication of estimated relative contributions of individual factors to periodontal disease outcome need to be described. Though the tobacco use was not associated for this study population, the suggested comprehensive approach/ model should also include cessation advice along with oral hygiene advice (considering negative impact of tobacco on general health and oral health, specially on periodontitis).

6. PLOS authors have the option to publish the peer review history of their article (what does this mean?). If published, this will include your full peer review and any attached files.

Reviewer #1: No

Reviewer #2: **Yes: **Masuma Pervin Mishu

---

## [Author Response · Author response to Decision Letter 0]

17 Feb 2021

Reply to reviewer#1 comments

1. Page 6, line 96. The exclusion criteria included previously receiving insulin prescriptions. Could you provide the percentage of diabetes patients who have received insulin prescriptions in Bangladesh? And, could you explain the reasons of such exclusion

- Response: According to the reviewer’s comments, We excluded patients who previously received insulin prescriptions as this may indicate more advanced stage of the disease. (Page 6, lines 118-119). The percentage of patients who have received insulin prescription in Bangladesh was less than 7.3% to 14.0%. (page 27, lines 521-526).

2. Table 1. What the difference between ”all” and “overall”? It’s confusing.

- Response: In accordance with the reviewer’s comments, we have revised the manuscript to have consistency with the terms. The prevalence of total periodontal disease was calculated for all diabetes patients with any type of periodontal disease versus no disease. The prevalence for severity was calculated for mild, moderate and severe categories of diagnosed periodontal disease patients (Pages 14-16; Table 1).

3. Table 1-3. The sample size for testing any form of periodontal disease was 379, but the sample size for testing the severity of periodontal disease was 287. Please explain the reasons in Methods, and make the columns of the tables more understandable.

- Response: Number of subjects 379 refers all diabetes patients included in this study and 287 refers diabetes patients with any periodontal disease. Binary logistic regression analysis was run for 379 patients comparing patients with any periodontal disease categories versus patients without periodontal disease. To apply ordinal logistic regression analysis for severity of periodontal disease (mild, moderate and severe) with mild disease as the reference level, only the diabetes patients with diagnosis of periodontal disease were included in the analysis (n=287). Amendments were made in the Analysis sub-section (Page 11, lines 247-252), in the Methods section (Pages 18-19, Table 2), and in the columns of the tables to clarify the meanings (Page 20, Table 3).

4. For the general population of non-diabetic patients, poor oral hygiene habits are also an important risk factor for early occurrence of periodontal disease. In this study, oral health habits and glycemic control behaviors were put into a model at the same time in patients with diabetes mellitus. The results showed that both the glycemic control behaviors and the oral hygiene behaviors played an independent role in the development and severity of periodontal disease in diabetic patients. Could you further analyze the interaction between these two, and stratify by oral hygiene behavior to see if poor glycemic control is associated with much higher risks of having any forms or severity of periodontal disease among patients with poor oral hygiene?

- Response: According to the reviewer’s concern, we have performed a stratified analysis according to oral hygiene behaviours. The results showed that patients with poor or fair oral hygiene practices with poor control of diabetes are associated with increased risk of any and severe periodontal disease. Table 4 was added to show the results. The methods, results and discussions corresponding of these results were added in the Abstract (Page 3, lines 46-48), Methods (Page 12, lines 264-266), Results (Page 21, lines 373-387), and Discussion sections (Page 25, lines 463-470). 

Reply to reviewer #2 comments

1. Abstract: Introduction: The second sentence of the Introduction needs to be simplified, as it is difficult to understand in the current version. The justification of the second objective “) the relative contributions of selected factors to periodontal disease outcome in type 2 diabetes patients.” should be added

- Response: According to the reviewer’s suggestion, the second sentence of the introduction has been clarified and the second objective has been justified (Page 2, lines 19-25)

2. Abstract: Results: The Demographics of the participants need to be added (% of male, female, mean age (Standard deviation). The severity % is not clear in the sentence: “The prevalence (75.7%) and severity (35.1%) of periodontal disease were high in type 2 diabetes patients.” The last sentence of the Result “Diabetes conditions and oral hygiene practices explained 10.9% and 7.3% of the variance in severe periodontal disease.” The “severe periodontal disease” is not clear.

- Response: According to the reviewer’s comments, the demographics characteristics of the participants were added. The second sentence was amended to clearly address the prevalence of any form of periodontal disease and severe form of periodontal disease. The sentence was amended to elaborate percentage coverage of variance by modelling of severe or moderate forms versus mild form of periodontal disease. (Page 2, lines 36-38; page 3, lines 41-48)

3. Introduction: 

The last sentence of Page 4 (line 71, 72) Association of periodontal health with self-care practice, healthy diet and physical activity has been shown as a gap in the literature. The potential undying mechanism of these associations need to be indicated to make the rational for investigating this associations. These factors seems to be directly related with the control of diabetes, which can have impact on periodontal health (similar is the case for Discussion page 22, line 375 and 376. The other potential explanation should be added)

Another point, why it is important to estimate the relative contributions of individual factors to periodontal disease outcome need to be justified (clinical practice related importance etc.)

- Response: According to the reviewer’s concern, the rationale to estimate the relative contributions of individual factors and the potential underlying mechanism of the association between self-care practices and the risk of periodontal disease were added in the Introduction section. (Page 5, lines 81-95; Page 22, lines 404-406)

4. Results: The demographic information of the study population need to be included in the first paragraph.

- Response: As suggested, the demographic information of the study population was added in the first paragraph (Page 13, lines 282-288). 

5. Discussion: The clinical implication of estimated relative contributions of individual factors to periodontal disease outcome need to be described. Though the tobacco use was not associated for this study population, the suggested comprehensive approach/ model should also include cessation advice along with oral hygiene advice (considering negative impact of tobacco on general health and oral health, specially on periodontitis).

- Response: According to the reviewer’s comments, the clinical implications of estimated relative contributions of individual variables to the outcome of periodontal disease were further addressed (Page 24, lines 456-461). The issue of tobacco use was also discussed (Page 25, lines 472-480).

---

## [Decision Letter · Decision Letter 1]

10 Mar 2021

Relationship between diabetes self-care practices and control of periodontal disease among type 2 diabetes patients in Bangladesh

PONE-D-20-36828R1

Dear Dr. Nakamura,

We’re pleased to inform you that your manuscript has been judged scientifically suitable for publication and will be formally accepted for publication once it meets all outstanding technical requirements.

Kind regards,

Toshiyuki Ojima, M.D., Ph.D

Academic Editor

PLOS ONE

Additional Editor Comments (optional):

Reviewers' comments:

Reviewer's Responses to Questions

**Comments to the Author**

1. If the authors have adequately addressed your comments raised in a previous round of review and you feel that this manuscript is now acceptable for publication, you may indicate that here to bypass the “Comments to the Author” section, enter your conflict of interest statement in the “Confidential to Editor” section, and submit your "Accept" recommendation.

Reviewer #1: All comments have been addressed

Reviewer #2: All comments have been addressed

2. Is the manuscript technically sound, and do the data support the conclusions?

Reviewer #1: Yes

Reviewer #2: Yes

3. Has the statistical analysis been performed appropriately and rigorously? 

Reviewer #1: Yes

Reviewer #2: I Don't Know

4. Have the authors made all data underlying the findings in their manuscript fully available?

Reviewer #1: Yes

Reviewer #2: Yes

5. Is the manuscript presented in an intelligible fashion and written in standard English?

Reviewer #1: Yes

Reviewer #2: Yes

6. Review Comments to the Author

Reviewer #1: (No Response)

Reviewer #2: Thanks for addressing the comments. Finally, the manuscript need a careful proofreading. For instance, the last sentence in page 24 (line 496- 461) need to be checked for grammatical correction.

7. PLOS authors have the option to publish the peer review history of their article (what does this mean?). If published, this will include your full peer review and any attached files.

Reviewer #1: No

Reviewer #2: **Yes: **Dr. Masuma Pervin Mishu

---

## [Editor Report · Acceptance letter]

29 Mar 2021

PONE-D-20-36828R1 

Relationship between diabetes self-care practices and control of periodontal disease among type2 diabetes patients in Bangladesh 

Dear Dr. Nakamura:

I'm pleased to inform you that your manuscript has been deemed suitable for publication in PLOS ONE. Congratulations! Your manuscript is now with our production department. 

Kind regards, 

on behalf of

Dr. Toshiyuki Ojima 

Academic Editor

PLOS ONE